# The Effects of Colour Content and Cumulative Area of Outdoor Advertisement Billboards on the Visual Quality of Urban Streets

Mastura Adam [1], Ammar Al-Sharaa [1], Norafida Ab Ghafar [1,*], Riyadh Mundher [2], Shamsul Abu Bakar [2] and Ameer Alhasan [3]

1   Department of Architecture, Faculty of Built Environment, Universiti Malaya, Kuala Lumpur 50603, Malaysia
2   Department of Landscape Architecture, Faculty of Design and Architecture, Universiti Putra Malaysia, Serdang 43400, Malaysia
3   Department of Computer Techniques Engineering, Dijlah University College, Baghdad 00964, Iraq
*   Correspondence: norafida@um.edu.my

**Abstract:** Visual comfort has a critical effect that significantly influences public appreciation of urban environments. Although colour is an integral part of billboard design, little empirical evidence exists to support some of the popularly held ideas about the effects of colour on task performance and human psychological wellbeing. Thus, attempting to set a threshold level of allowed undesirable visual stimuli in each urban setting is considered to be essential in achieving a satisfactory level of visual quality. Therefore, this research investigates the effects of colour content of outdoor advertisement billboards on the appreciation of urban scenes by the public. This research utilises pictorial survey, R.G.B bivariate histogram technique, and an areal cumulative analysis of a group of collected pictures within one of Kuala Lumpur's high streets. Results of the pictorial survey are cross analysed against the results of the pictorial RGB content analysis and pictorial outdoor advertisement (OA) cumulative areal analysis to indicated a strong correlation between environmental colour content, OAs' cumulative area, and visual comfort. The study suggests that the lack of guidelines and regulations of the color content of outdoor billboard advertisement design could potentially be detrimental for the public's appreciation of urban environments. Future research initiatives are encouraged to develop a visual quality assessment framework that contributes to the image and identity of the city of Kuala Lumpur.

**Keywords:** outdoor advertisement; colour content; urban landscape; urban street; visual comfort; visual pollution; visual quality; urban streets

## 1. Introduction

Visual pollution can be defined as a collection of elements which give the public a feeling of discomfort towards a particular scene. This feeling can be caused by an increase of certain objects in the environment [1,2], or the degradation of the elements, and objects, that are observed in the landscape [3,4]. It also can be influenced by other scenic objects that are typically seen in the urban environment such as outdoor billboard advertisements and signages [5,6], or industrial facilities [7], especially when they are out of the visual context. Concerns regarding visual pollution caused by outdoor advertisements started as early as the 1940s and 1950s stemming from the growth of automotive transportation and the construction of the interstate highway system in the United States. This interest grew into a public and a political movement with goals to regulate the increase of outdoor billboards resulting in the Highway Beautification Act in 1965 [8].

However, in the urban environments of today, advertisement billboards and signages are becoming more widespread due to the progression of digital and printing techniques that allowed for larger, more affordable, and more durable advertisement boards which

allow for the longest possible display time [9]. These issues can affect the physical and social quality of urban space, when an extreme use of advertisement billboards takes place [10]. Previous research suggested that billboards were in some cases hazardous to drivers' safety due to the potential risk of high distraction levels [11,12]. Further research suggested that the abundance of advertisement billboards may cause some unfavourable long-term effects on the public, such as risking the public's future mental wellbeing. The same estimation was mentioned by Thomas [13], who deemed advertisement billboards as a pollution source that participates in causing psychological disorders. These disorders may include psychological disorders such as dysphoria, diabetogenic eating behaviours, and compulsive buying [14].

The fast-paced increase in the outdoor advertisement boards in the city centre can also be correlated with the economic pressure faced by the municipal authorities to increase their annual financial yields [7]. Payments imposed to procure the permits for outdoor advertising can be a highly lucrative source of income that may play a critical role in lifting some of the local authorities' financial burdens [15]. Outdoor advertising indicates a struggle faced by local authorities in balancing the benefits of economic revenues for the urban areas and fulfilling both the aesthetic [16,17], and public health obligations towards the communities, on the other hand [2].

In the Malaysian setting, the Malaysia Investment Development Authority indicates that there are 179 local municipal offices in Malaysia, and it is their responsibility to approve signboard licenses. It is worth mentioning that the requisites of obtaining an outdoor advertisement billboard's license may vary with regard to the stipulations set by a particular local authority [6]. The urgency of investigating visual pollution's potential adverse effects on the perceived environmental quality has been discussed in previous research [18]; however, little research has discussed the effects of colour composition on urban streets' public appreciation [19]. Therefore, the objective of this study was to investigate the effect of the urban environmental colour content on visual comfort.

*Research Background*

The visual experience in urban spaces is essential for developing a fulfilling user experience. The manifestation of a positive visual image of urban space will foster the creation of positive impressions, therefore inspiring the community to have a favourable reaction to its urban environment. In addition, to build effective urban streets visual quality assessment tools, it is vital to increase and encourage community engagement in the process of controlling the visual quality of urban areas by assessing the visual quality of street scenes [20]. Tang and Long [21] mentioned that when streets are equipped with an adequate form or design they have the potential to increase physical functionality and may trigger an emotional appeal. Furthermore, the social meaning and symbolic image of a street space may be associated with its perceived feeling [22].

Colour can form the basis of visual comfort in building interiors as well as urban and semi urban settings. It can also contribute to the effectiveness of direction signs and orientation systems [23], thus contributing to safety [24], effectiveness [25], well-being [26], and spatial recognition of wayfarers [27]. Furthermore, colour contrast and intensities may help to differentiate various spatial functions, signalise directions to destinations, reference intersections, movement paths, destinations, and information delivery points [28].

Outdoor advertisements' colour and the appreciation of the urban landscape have been mentioned by Abdelhamid [29] as contributing factors alongside other prominently studied factors such as the number of visual stimuli. Colour as a factor influencing the perception of visually cluttered urban areas has been studied with a focus on urban landscapes' heterogeneity [30], and its potential role in amplifying feelings and emotions. Thus, colour pollution can cause an adverse visual response in the urban landscape due to the uneven application of colour. Therefore, human perception of colour can be a source of unpleasantness and visual discomfort toward the urban scene at the extreme [31]. The effect of commercial street board arrangements on environmental aesthetic, in particular

on commercial streets, has a significant impact on people's appreciation, as highlighted by Cubukcu et al., [32]. Pedestrians meet interesting and varied experiences as they walk along the street facades at 5 km an hour speed. Visual contact is close and personal as the pedestrian is able to take in everything around them. Thus, the rhythm of visual qualities offered along the street is crucial to the richness of the pedestrian experience: the number of doors, windows, niches, columns, shop windows, display details, signages, and decorations is significant [33]. In addition, walking becomes even more appealing if the details and displays along the way are carefully designed, and engage human senses in seeing, hearing, smelling, and touching. Visual comfort exists when the perceptual faculties in the human brain can operate without interference [34]. When there is no inhibition of perception, the basic functions of the eyes, such as vision, speed, and contrast sensitivity, are optimised. This optimisation of the basic perceptual functions is very important while perusing optimal working conditions [35]. Computer-based models have become an important element of sensitive decision-making processes [5,36], especially when a multi-criterion evaluation procedure is applied. Developing a visual pollution assessment technique has gained some attention in recent years [37], amongst which is the employment of mathematical decision-making approaches such as the analytical hierarchy process (AHP). Moreover, Chimlewlski [5] has implemented an intervisibility analysis using Geographic Information System (GIS) techniques as a measure of quantifying visual pollution perceived thresholds coupled with a public survey. Furthermore, Bakar et al., [6] have suggested that the cumulative area of the pollutants in ratio to the total viewed scene can play a role in shaping perceptual responses towards urban scenes. Chmielewski [38] has attempted to take visual pollution analysis to a quasi-third dimension analysis or what was referred to as 2.5D by importing data from laser scanning of a scene, and through digital analysis of the data a "third" dimension can be extracted and then mapped on-site for further analysis. Similarly, Chmielewski [39] has employed a 3D Isovist technique as a visual impact assessment of billboards in urban areas. Moreover, Wakil et al. [40] have employed an assembly of open-source geospatial tools to assess and map urban visual pollution. Further research articles have focused on the development and establishment of reliable measurement tools that could enable local authorities to capitalise on the process of regulating advertisements and perhaps devise taxation plans based on the potential harm these outdoor advertisements may do to the public's wellbeing [41]. Tang and Long [21] have identified five categories of approaches that have been employed to assess the scenic quality of urban streets, namely: subjective perception assessment, which includes the incorporation of in depth interviews [42], or in-person survey questionnaires [43]; systematic observation and rating of streetscapes, which includes the rating of site surveying-based imagery through videotaping or street view pictures [20,44]; physiological monitoring which includes the incorporation of electroencephalograms (EEG) [45] or eye tracking metrics [46]; laboratory experiments, which include the incorporation of virtual reality (VR) [36], or augmented reality (AR) [47]; and computer-assisted auditing and evaluation which includes the incorporation of GIS data, remote sensing, and image processing [48]. Other taxonomies regarding the assessment of human-environmental interaction have classified investigation approaches based on the focus of the study (i.e., spatial, perceptual, etc.), and objective expert approaches or subjective expertise approaches [49,50].

This study proposes a user-centred assessment approach with a mixture between the "systematic observation and rating of streetscapes approach" through the rating of the perceived pictorial visual comfort using a Likert scale technique and computer-assisted auditing and evaluation through an image processing technique.

Hence, this study attempts to assess the visual comfort of urban streets through the incorporation of the streets' users as the basis of the assessment in terms of rating the street's scenic comfort as well as involving two pictorial analysis techniques, namely the areal cumulative analysis of OA billboards as well as the pictorial R, G, B bivariate histogram. Consequently, the results may aid in the identification of critical aspects and factors of the perceived visual comfort of urban facades.

## 2. Materials and Methods

### 2.1. Description of the Study Site

Kuala Lumpur (KL), the capital of Malaysia, is considered one of the most important financial centres in Southeast Asia and a very attractive tourist destination [51]. The selected area of the study was located within a section of Jalan Tuanku Abdul Rahman (JTAR) starting from the junction of Jalan Dang Wangi towards the intersection of Jalan Tun Perak. This stretch of the street was selected due to its historical importance and being perceived as one of the most successful shopping areas in the Klang Valley (Figure 1).

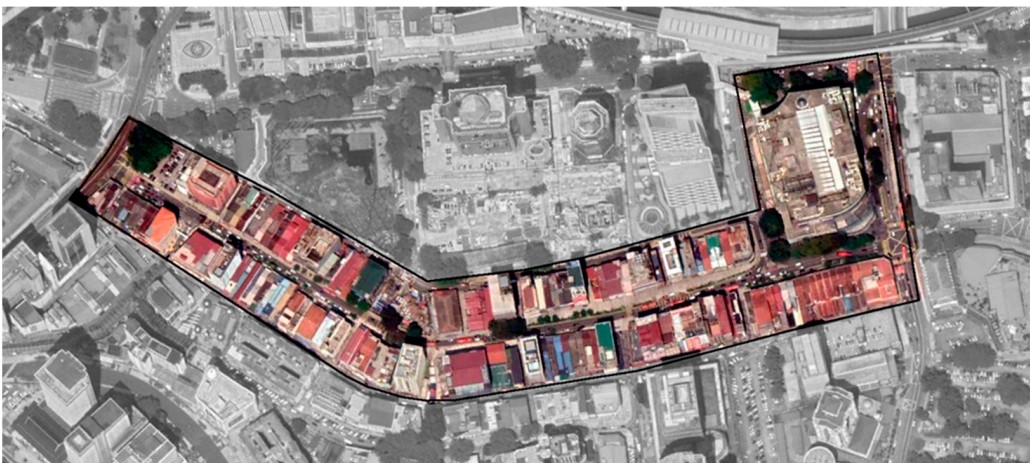

**Figure 1.** Location map of the study area in Kuala Lumpur (Image source: Google Earth Pro).

According to Mahalingam [52], the JTAR area is well known for textile trading, and the shop lots are characterised as the "most sought-after real estate" amongst shopping districts in KL. That resulted consequently in the buildings' facades in JTAR being scattered with different sizes of signage and billboards (Figure 2).

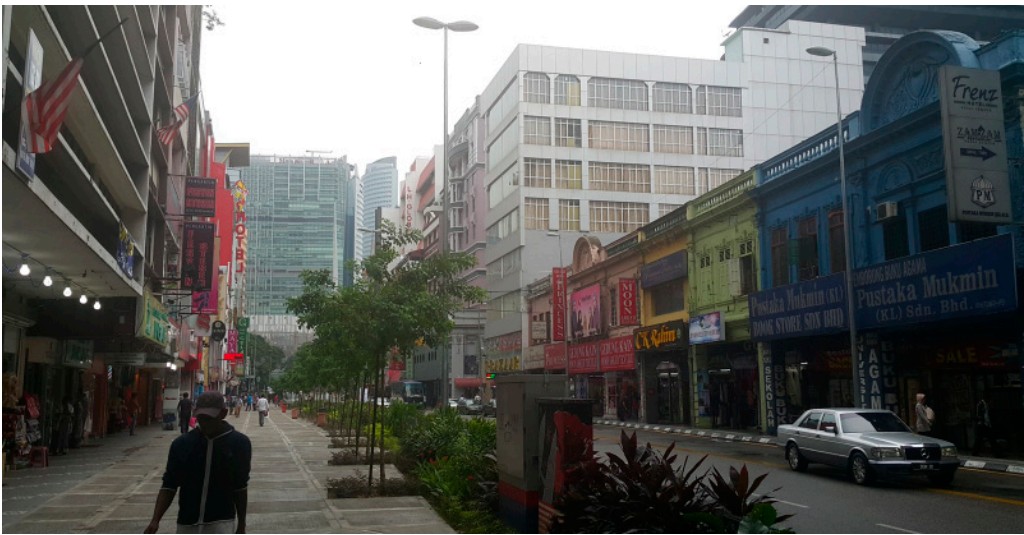

**Figure 2.** The billboard advertisement at the historical section of JTAR street.

JTAR gained a reputation as an attraction site in the city of Kuala Lumpur and has been further strengthened by the authorities' plan to limit vehicular movement. The authorities plan to encourage a pedestrian-friendly environment for the shoppers and provide the opportunity to the local municipal authorities to further develop the existing

urban landscape in the district. A recent survey by the local authority found that 65% supported the idea [53], engaging more than 3000 respondents.

### 2.2. Data Collection

The study methods can be divided into two main sections, data collection and data analysis. The data were mainly collected via two means, namely site photography and a public survey. On the other hand, the analysis stage comprised three processes: areal cumulative analysis, RGB content analysis, and statistical analysis, as seen in Figure 3.

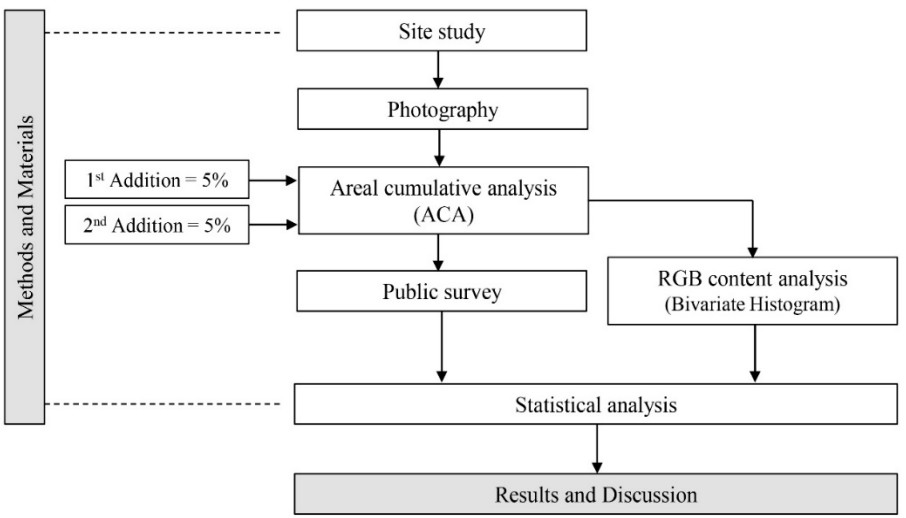

**Figure 3.** The methodological flowchart of the study.

This study relied on analysing a set of collecting photographs through site visits and the application of two pictorial analysis techniques, namely RGB based histogram, which is an image colour segmentation technique [54], and billboard cumulative areal analysis technique [6]. Then, the pictorial analysis results were tested against the perceptual data gathered through a survey of responses towards a displayed set of analysed scenes, in an attempt to expand our understanding of the previously collected results. It was not concluded whether there is a perceptual threshold of visual pollution and, if one was to be detected, where it is or how someone could measure it.

### 2.3. Field Work

Figure 4 shows the locations of the images used to investigate the extent to which the public can tolerate the ever-growing visual contamination. The images were acquired in 20 different locations on each side of the JTAR sideways (10 locations on each side). The distance to each location was about 70 m and covered about 700 m from the road.

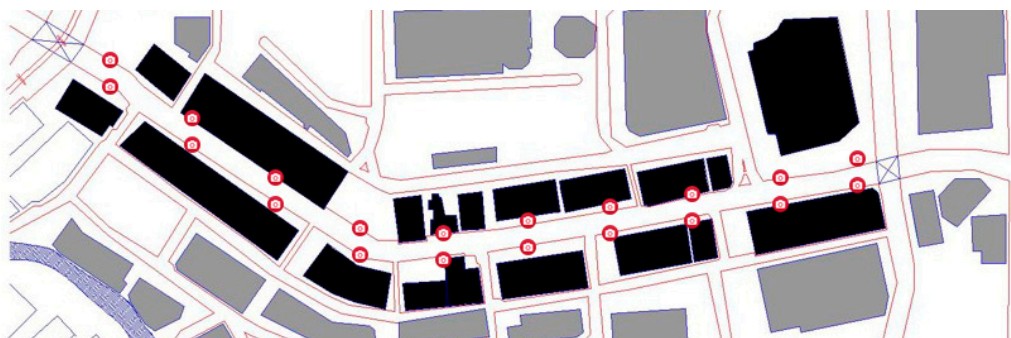

**Figure 4.** Locations of the collected pictures at JTAR before the sorting and elimination process.

Three images were taken in each position covering the street view and the camera lens was tilted at a pedestrian-level angle to ensure the best collection of scenes that best reflected the presence of foreground elements on the street, in the centre, and in the background. A total of 60 images were collected on the site.

The images went through a sorting process where some of the images were removed from the main list due to the presence of unnecessary objects such as large trees or buses that have blocked the street view. From this process, a final selection of 14 images in total from 14 different locations was selected to simulate higher levels of visual pollution and survey.

### *2.4. Data Analysis Techniques*

#### 2.4.1. Pictorial Analysis

There are two pictorial analysis procedures employed in this study to measure two factors, namely the RGB analysis using the bivariate histogram analysis technique as described in Section 2.4.2 and pictorial areal analysis of outdoor advertisement billboards using areal cumulative analysis, explained in detail in Section 2.4.3. Performing a cross-analysis between subjects' responses and these two factors will provide a multivariate approach to correlating visual pollution elements with the public's urban appreciation.

#### 2.4.2. RGB Analysis Using RGB Histogram

A pictorial colour content analysis method was incorporated using RGB content calculation, starting by reducing the image size to 250 × 250 pixels due to limited computational power. An RGB histogram was created that demonstrates the colour frequency in any of the 42 selected pictures, as seen in (Figure 5). Furthermore, the frequencies of the pixels in CSV extensions were extracted. Each colour ranges in intensity from 0 to 255 with zero being the least intense and 255 being the most intense.

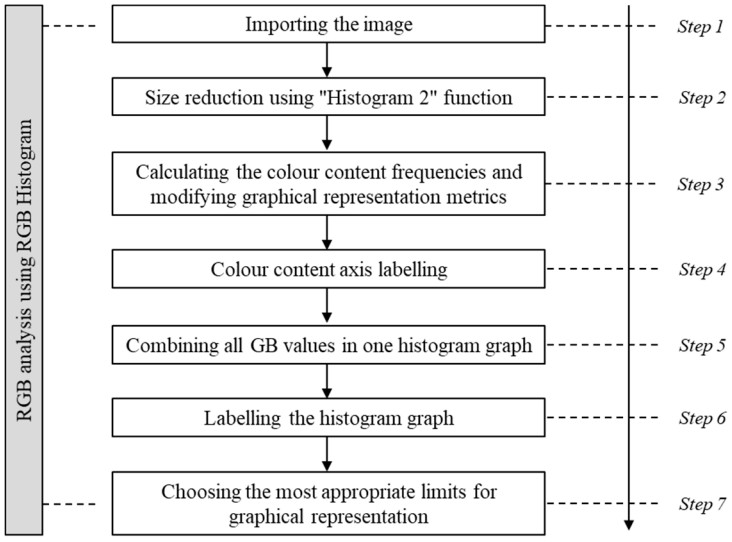

**Figure 5.** The steps needed to perform the RGB bivariate histogram analysis and display.

Each picture contains 65,025 pixels with each pixel having a value for red, green, and blue. The steps needed to perform the RGB bivariate histogram analysis are shown in Figure 5. In this technique, the outcome can be exported in multiple CSV formats including the colour content data, jpeg formatted graphs representing two colours juxtaposed to each other, and a graphical representation of the colour content measured by frequencies and intensity (Figures 6 and 7).

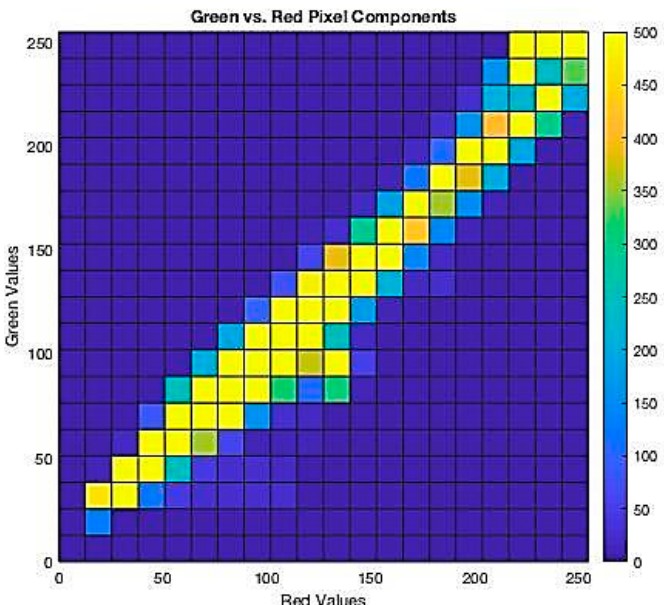

**Figure 6.** Green vs. red pixel components bivariate histogram that illustrates the colour frequencies and intensity.

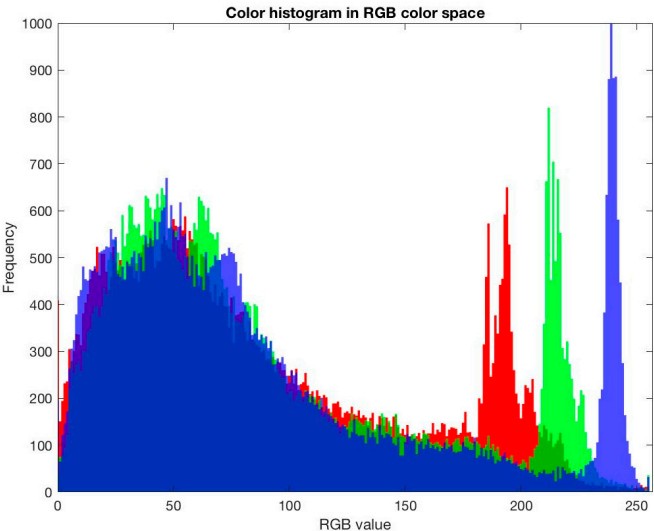

**Figure 7.** R.G.B. histogram that illustrates the colour frequency and intensity by RGB value.

The frequencies in this scenario represent the no. of pixels concerning a specific RGB value. In addition to the content and frequencies of each colour, a histogram of all colours combined was created (Figure 7). Statistical analysis was conducted to gauge the effectiveness of each colour on visual perception of the scenes on the mean values of the responses for each picture.

### 2.4.3. Areal Cumulative Analysis (ACA)

The images were further analysed to identify the existing level of visual pollution caused by the advertisement boards. By using Autodesk AutoCAD, the images were 'Rastered' and the visible area of these advertisement boards was digitised to calculate its cumulative amount of exposure. Figure 8; Table 1 reveals that the total amount of exposures for each scene varies from 0.82% (lowest) to 21.34% (highest) of the total scene area. The images then were assigned specific alphabetical codes for further image manipulation.

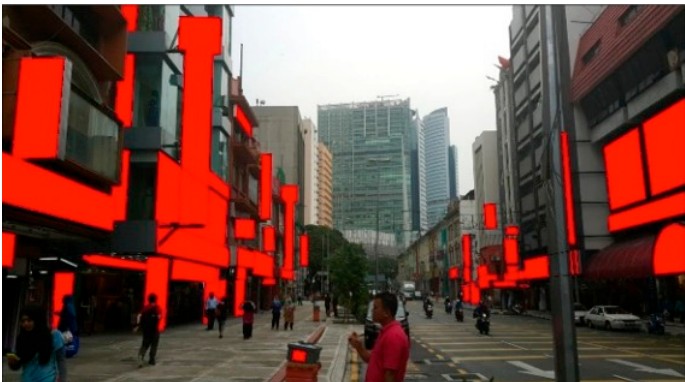

**Figure 8.** An example of area cumulative analysis for one of the scenes shows a scene in AutoCAD's digital environment (Code: A) the existing level of advertisement board's exposure (%) = 16.27% of the total pictorial area.

**Table 1.** Existing level of pollution exposure in %.

| Picture | Code | Existing Level of Exposure % | Picture | Code | Existing Level of Exposure % |
|---------|------|------------------------------|---------|------|------------------------------|
| 1 | A | 16.27 | 8 | H | 14.66 |
| 2 | B | 17.28 | 9 | I | 1.42 |
| 3 | C | 11.79 | 10 | J | 2.16 |
| 4 | D | 10.65 | 11 | K | 0.82 |
| 5 | E | 11.08 | 12 | L | 7.00 |
| 6 | F | 21.34 | 13 | M | 6.85 |
| 7 | G | 10.13 | 14 | N | 3.68 |

In addition, larger areas of advertisement boards were added to the original images to increase the pollution exposure. Each image received two levels of pollution increment (1st = +5% and 2nd = additional +5%) to the total pictorial area. For each level of addition, a new code was assigned based on the scenes' original coding. The final number of scenes (n = 42) were arranged in a random sequence with the highest level of cumulative pollutants area of 31.21.

Photos were then analysed (including both original and simulated levels of exposure) via the bivariate histogram technique. A sample of an original picture (code F), as well as its modified versions with additions made of approximately 5% increments, and the respective RGB histogram can be seen in Figure 9.

The final images were then employed in a photo survey in which the photos were displayed and the respondents were asked to rate each of the images using a Likert scale rating: 1-highly pleasant, 2-pleasant, 3-moderately unpleasant, 4-unpleasant, and 5-highly unpleasant. The ordered scale allowed the respondents to choose the best answer that aligns with their views easily. Another indicator of potential importance that was not discussed previously was the number of ads in the pectoral area. Therefore, this study investigates a set of factors such as area of pollutants, number of billboards, and pictorial colour content as a representation of the environmental colour content.

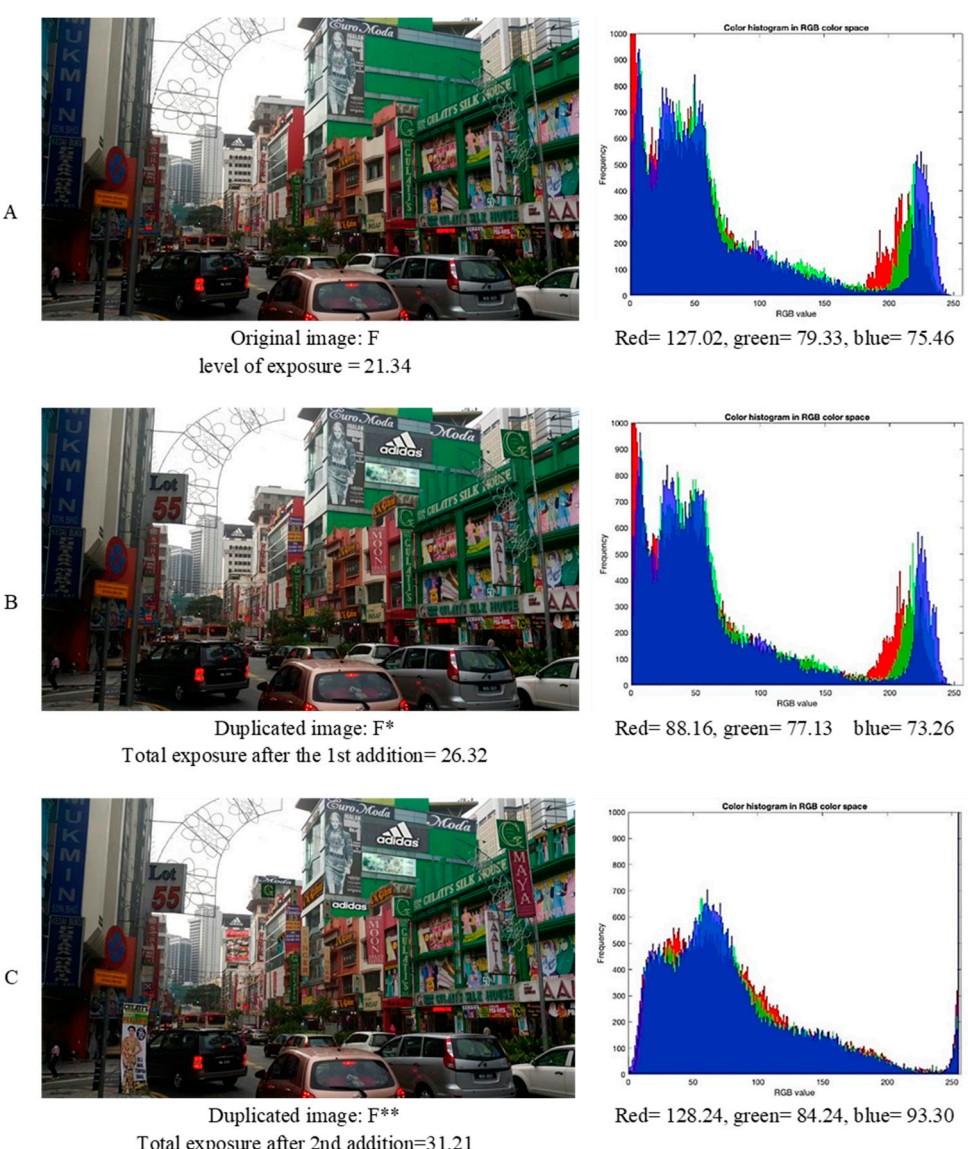

Original image: F
level of exposure = 21.34

Red= 127.02, green= 79.33, blue= 75.46

Duplicated image: F*
Total exposure after the 1st addition= 26.32

Red= 88.16, green= 77.13    blue= 73.26

Duplicated image: F**
Total exposure after 2nd addition=31.21

Red= 128.24, green= 84.24, blue= 93.30

**Figure 9.** A sample original picture (Code F), the additions made of approximately 5% increments, and the respective RGB histogram. Whereby (**A**) represents the original photo, (**B**) represents the photo after 1st addition, and (**C**) represents the photo after 2nd addition.

## 2.5. Sampling Design and Analysis

A purposive sampling process took place by inviting a group of subjects with design and architecture education backgrounds from multiple levels due to their familiarity with the subject. The survey was divided into two main sections, the demographic and visual simulation sections. The questions included in the demographic sections were gender, age, place of residence, level of education, frequency of visits, and existing knowledge. These factors were investigated as independent variables to examine the factors that influence appreciation for public spaces. The data resulting from registering people's preferences towards certain scenes were then statistically tested against the cumulative visible area of billboards from the human perspective, as well as the pictorial RGB content. A detailed description of the analysis tools used in this study is presented in the next section. A set of statistical techniques were employed using IBM SPSS25 in this study. A normality test was conducted to investigate the data distribution in order to determine the suitability of the parametric test. A Pearson correlation test was then adopted to investigate the relationship between different levels of visual stimulation represented by the pictorial cumulated area with the general collected results. Additionally, an analysis of variances ANOVA test took

place to analyse the differences in responses amongst different demographic groups after testing the data for normality.

## 3. Results

### 3.1. Respondents' Demographic Analysis and Normality of Distribution

Respondents volunteered for our survey (n = 59). The respondents' age varied from 20 years to 34 years old, and a majority of them were females (61%; n = 36) and the rest were males (39%; n = 23). The number of respondents who hailed from urban areas was (52.5%; n = 31) while (48%; n = 28) were from suburban/rural areas. Malaysian students made up the majority of the group (79.7%; n = 37) while the rest (20.3%; n = 12) were foreign students. The initial descriptive statistical analysis indicated that the respondents comprised a total of 22 respondents with the academic achievement of an undergraduate degree (bachelor's degree or higher) accounting for 37.3%, and 37 respondents (62.7%) reported not having obtained an academic tertiary education. For the visual simulation survey, Cronbach's alphas tested for the 42 items shows a higher degree of reliability (42 items; $\alpha = 0.908$).

Data normality was investigated through the kurtosis, skewness, as well graphical investigation of boxplots, and Q-Q plots. The results indicated kurtosis values between 1 and −1, with kurtosis levels between 2 and −2, for all the measurements. Therefore, the data are considered moderately symmetrical and the level of normality is considered acceptable [55]. The data's symmetry and normality of distribution allows for the employment of parametric tests, such as Pearson correlation and ANOVA tests.

### 3.2. Areal Cumulative Analysis Results

Analysis of the cumulative areas of the pollutants and the effects of pollution levels on subjects' responses were carried out. A summary of the areal analysis results represented by percentages of the total pictorial area can be found in Table 2.

**Table 2.** Summary of the pictures of areal visual pollution levels represented by the pictorial area and the number of billboard advertisements.

| | Existing | | | After 1st Addition | | | After 2nd Addition | |
|---|---|---|---|---|---|---|---|---|
| Code | Level of Exposure % | No. of OAs | Code * | Level of Exposure % | No. of OAs | Code ** | Level of Exposure % | No. of OAs |
| A | 16.27 | 51 | A * | 21.23 | 54 | A ** | 26.09 | 62 |
| B | 17.28 | 36 | B * | 22.39 | 42 | B ** | 27.43 | 54 |
| C | 11.79 | 45 | C * | 16.69 | 53 | C ** | 21.59 | 59 |
| D | 10.65 | 45 | D * | 15.55 | 51 | D ** | 20.56 | 57 |
| E | 11.08 | 43 | E * | 16.08 | 49 | E ** | 20.98 | 59 |
| F | 21.34 | 54 | F * | 26.32 | 62 | F ** | 31.21 | 78 |
| G | 10.13 | 26 | G * | 15.26 | 30 | G ** | 20.38 | 46 |
| H | 14.66 | 34 | H * | 19.67 | 39 | H ** | 24.64 | 31 |
| I | 1.42 | 10 | I * | 6.45 | 18 | I ** | 11.59 | 27 |
| J | 2.16 | 8 | J * | 7.02 | 15 | J ** | 12.03 | 19 |
| K | 0.82 | 6 | K * | 5.79 | 15 | K ** | 10.67 | 21 |
| L | 7.00 | 19 | L * | 12.02 | 23 | L ** | 17.15 | 26 |
| M | 6.85 | 14 | M * | 11.87 | 18 | M ** | 16.94 | 25 |
| N | 3.68 | 12 | N * | 8.65 | 15 | N ** | 13.76 | 18 |

Note: the "*" sign indicates a 5% areal addition to the original photo. Note: the "**" sign indicates a 10% areal addition to the original photo (second addition of 5%).

### 3.3. Cross-Sectional Analysis between RGB Content and Subjects' Responses

Averaging the pictorial content of each colour in each pixilated pictorial region resulted in a specific mean value assigned to each picture (Table 3).

**Table 3.** Summary of colour content means for each picture.

| Code | Green | Blue | Red | Code | Green | Blue | Red |
|---|---|---|---|---|---|---|---|
| I ** | 81.26 | 75.88 | 97.26 | M * | 91.12 | 91.06 | 88.97 |
| J | 108.12 | 94.04 | 83.34 | J ** | 88.72 | 91.62 | 87.93 |
| N | 111.30 | 106.92 | 71.93 | A | 71.41 | 66.66 | 70.70 |
| K * | 113.36 | 106.32 | 83.80 | K ** | 77.97 | 77.19 | 80.62 |
| A * | 95.94 | 94.97 | 103.11 | M | 90.12 | 79.76 | 96.50 |
| F ** | 84.24 | 93.30 | 128.24 | F | 79.33 | 75.46 | 127.02 |
| B * | 92.60 | 93.37 | 112.90 | M ** | 98.96 | 96.33 | 98.60 |
| L ** | 95.94 | 91.42 | 89.22 | C ** | 73.76 | 90.60 | 99.77 |
| H | 97.03 | 104.39 | 73.26 | N ** | 91.37 | 91.06 | 92.00 |
| E ** | 88.90 | 89.83 | 96.50 | F * | 77.13 | 73.26 | 88.16 |
| D * | 117.23 | 102.64 | 72.86 | E * | 87.05 | 88.37 | 84.48 |
| N * | 133.10 | 116.44 | 69.31 | G * | 84.67 | 77.94 | 89.65 |
| J * | 95.72 | 72.06 | 95.90 | I * | 84.04 | 76.06 | 85.55 |
| B | 81.23 | 79.87 | 84.34 | D | 103.20 | 109.75 | 73.85 |
| D ** | 81.48 | 67.96 | 111.22 | L * | 80.68 | 76.68 | 86.52 |
| C | 84.49 | 79.17 | 87.19 | B ** | 81.48 | 71.41 | 90.08 |
| L | 129.10 | 116.01 | 70.84 | H ** | 88.90 | 89.83 | 85.55 |
| H * | 73.28 | 80.23 | 85.77 | A ** | 72.76 | 76.06 | 117.16 |
| E | 84.00 | 87.41 | 87.94 | G ** | 92.54 | 92.20 | 129.57 |
| G | 91.12 | 74.88 | 88.97 | C * | 71.12 | 77.29 | 90.64 |
| K | 92.40 | 94.97 | 83.34 | I | 132.54 | 118.535 | 71.76 |

Note: the "*" sign indicates a 5% areal addition to the original photo. Note: the "**" sign indicates a 10% areal addition to the original photo (second addition of 5%).

A Pearson correlation test was performed to analyse the relationship between the cumulative area, the pictorial RGB content, and the overall response. The analysis showed a correlation between the cumulative area (level of exposure), the pictorial RGB content, and the overall response, as seen in Table 4.

**Table 4.** The correlation between the collected perceptual response as a dependent variable and the level of pollution and the RGB values as independent variables.

| | Correlations | Level of Exposure | Green | Blue | Red | Response |
|---|---|---|---|---|---|---|
| Level of exposure | Pearson Correlation<br>Sig. (2-tailed) | 1 | −0.541 **<br>0.000 | −0.365 *<br>0.018 | 0.594 **<br>0.000 | 0.697 **<br>0.000 |
| Green | Pearson Correlation<br>Sig. (2-tailed) | −0.541 **<br>0.000 | 1 | 0.855 **<br>0.000 | −0.443 **<br>0.003 | −0.565 **<br>0.000 |
| Blue | Pearson Correlation<br>Sig. (2-tailed) | −0.365 *<br>0.018 | 0.855 **<br>0.000 | 1 | −0.367 *<br>0.017 | −0.501 **<br>0.001 |
| Red | Pearson Correlation<br>Sig. (2-tailed) | 0.594 **<br>0.000 | −0.443 **<br>0.003 | −0.367 *<br>0.017 | 1 | 0.554 **<br>0.000 |
| Response | Pearson Correlation<br>Sig. (2-tailed) | 0.697 **<br>0.000 | −0.565 **<br>0.000 | −0.501 **<br>0.001 | 0.554 **<br>0.000 | 1 |

Note: *. Correlation is significant at the 0.05 level (2-tailed). Note: **. Correlation is significant at the 0.01 level (2-tailed).

A statistically significant correlation can be observed between the response and all the other variables, whereby the sign (2-tailed) values registered were 0.000 in all cases except for blue where it registered a 0.001. Furthermore, the results indicated an inverted correlation between green and blue values and the mean level of exposure, registering values of −0.365 and −0.541 respectively, indicating that with more exposure red became more present.

### 3.4. The Difference in Response to Visual Pollution Based on Demographic Group

An analysis of variances ANOVA-test was conducted to gauge the equality of variances of responses of different demographic groups towards pictorial colour content. The ANOVA-test was conducted to test five variables, namely, gender, age group, level of education, living environment, and frequency of travel to the city centre, for pictures grouped based on their level of exposure with approximately 5% increments, as seen in (Table 5).

**Table 5.** Pictorial group classification based on 5% increments.

| Group | OA Area Exposure % |
| --- | --- |
| 1 | 0–4.99% |
| 2 | 5–9.99% |
| 3 | 10–14.99% |
| 4 | 15–19.99% |
| 5 | 20–24.99% |
| 6 | 25%–above |

Analysing differences in response between gender groups indicated a significant difference in group 6 only with a $p$-value of 0.004, whereas male respondents seem to be more sensitive towards higher levels of pollution than females, registering a mean value of 3.9348 and a standard deviation of 0.47803 as opposed to their female counterparts, registering a mean value of 3.3472 and a standard deviation of 0.86453.

Grouping subjects into two sets of respondents, namely postgraduate education and undergraduate achieved education, testing differences in response based on their highest achieved education indicated no significant difference in responses across all the pictorial groups with $p$-values exceeding 0.05. Therefore, the null hypothesis of the equality of the variances cannot be rejected. Similarly, the test result registered no significant differences in responses to all the pictorial groups when subjects were grouped based on their living environments or their frequency of travel to the city centre, regardless of the differences registered in the mean response.

However, when the subjects were grouped into two age groups, 26 and above and 25 and below, the results indicated a significant difference with $p$-values of 0.023, 0.038, 0.004, and 0.046 in groups 1, 2, 3, and 4, respectively, which indicates that the null hypothesis of the variances being equal cannot be accepted. Alternatively, the analysis shows no significant differences in response to groups 5 and 6 where it seems like there were fewer variations in response to higher levels of pollution. The results showed that the older respondents seem to be more sensitive to lower levels of visual pollution represented by the first four groups, seen in Figure 10.

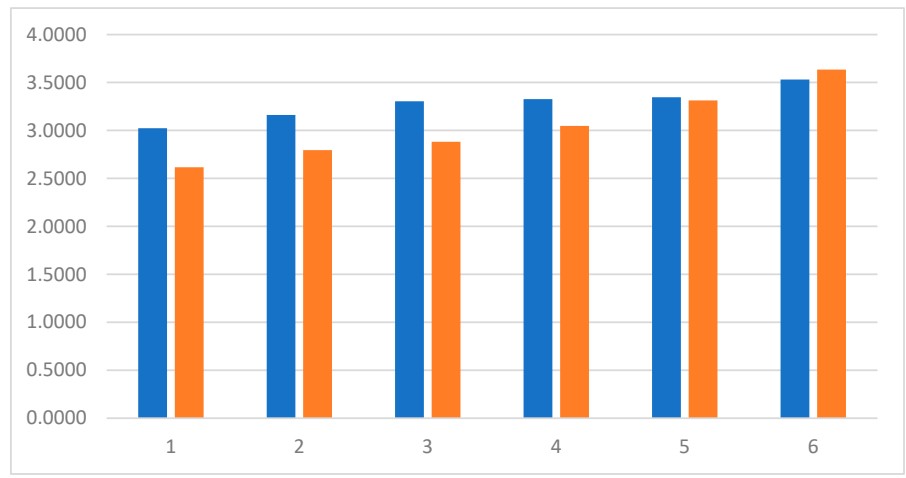

**Figure 10.** Difference in response between age groups.

## 4. Discussion

The survey questionnaire contained a question investigating the public's perception of how visually pleasing the urban environment is. An examination of the correlation test on the effect of colour on subjects' responses indicated that there was a statistically significant correlation between RGB colour content and response means. This finding is in line with Ahmed et al. [56] with regard to the effect of environmental colour on subjects' environmental perception. However, it is worth mentioning that the red colour content had a negative impact on subjects' appreciation, registering a positive Pearson correlation value of 0.560 and a sig. value of 0.000. On the contrary, both green and blue colours had the opposite effect, where it seems that blue and green had caused an increase in the level of scenic appreciation, registering negative Pearson correlation values of $-0.636$ and $-0.550$, respectively, and a sig. value of 0.000. This finding of the effect of colour content on subjects' appreciation is consistent with the findings of [1]. Furthermore, the results also indicate a significant correlation between the cumulative area and the mean response. This finding is consistent with Bakar et al., [6] with regard to the effect of cumulative area of outdoor advertisements on subjects' responses.

Moreover, the ANOVA test conducted to investigate the differences in responses amongst various demographic groups indicated no statistically significant differences in gender with regard to category 6 containing 25% and above the cumulative area of advertisement of the total pictorial area, registering a *p*-value of 0.004. This finding is consistent with Huang et al. [57] with regard to gender differences in terms of colour preferences. Furthermore, the ANOVA test indicated a significant difference in response amongst respondents from different age groups, whereby respondents above the age of 26 indicated a statistically significant difference in response towards the first four pictorial categories. However, no difference was observed in pictorial category number 5 and 6, which indicates that older adults were more sensitive than younger adults whereby they have registered higher levels of response towards lower levels of pollution. This finding is consistent with the findings of Lienemann et al. [58] concerning the variation in behavioural responses to advertisements in general.

The results of this study contribute to the area of development of evaluation methods for measuring the effects of visual pollution on the public's appreciation of urban areas. However, the findings of this study are not without limitations. Firstly, the findings are limited by the sample size; a larger sample size could produce a more generalizable set of findings. Furthermore, the study's respondents' age profile (20–34 years old) represents a narrow section of the population; a more diverse group of respondents may further highlight and perhaps add further insights into age group-based response variation. Secondly, the findings of this study are limited by the fact that we targeted a single area to be the case study. Further studies in which the same procedure can take place in a different environment could help to generalise the procedure as a procedure that can be used to evaluate the effects of visual pollution and colour content in other places within the Malaysian setting. Thirdly, this study employed a bivariate histogram image analysis technique to gauge the effects of colour content of urban scenes on the public's appreciation. This approach attempts to quantize the problem of visual pollution; however, there could be a qualitative aspect to the appreciation of urban scenes that is ignored by this study. Future studies are encouraged to take on the effects of visual pollution in general and colour content in specific by employing a qualitative technique to further the understanding of the effects of excessive outdoor advertisement on the public. The emergence of modern technologies such as the employment of virtual reality tours can be implemented in this regard to provide a better understanding of pedestrians' behaviour in these different urban settings. A holistic urban environmental visual assessment framework is lacking in the literature. Therefore, a conjoined analysis framework of the environmental factors such as colour content, environmental morphological features such as enclosure-exposure, and how it affects different demographic subsets of society represents a frontier of research in the science of wellbeing and urban design.

## 5. Conclusions

In summary, by analysing the RGB pictorial data shown to be correlated with the polluted area represented by the pictorial cumulative analysis, we can conclude that the environmental colour content represented by the RGB analysis was effective in the process of investigating the urban environmental conditions and how it could influence the public's appreciation. The study has found that red colour content seems to be negatively affecting our sample's appreciation of the urban scene. Therefore, we see this as a call for regulating the content of outdoor advertisements due to their potentially detrimental effects on the public's appreciation. The simulated higher level of pollution triggered negative responses by the study's respondents that have manifested in lower appreciation mean values.

This study is motivated by the need to promote the physical and psychological well-being of the residents of the city of Kuala Lumpur. Although several studies were performed to examine several other environmental concerns such as light, noise, air, and water pollution, the effect of visual quality on the public's appreciation of Kuala Lumpur's urban environment is still not being thoroughly investigated. The effect of cumulative area of outdoor advertisement billboards on visual colour content of urban scenes has shown to have several implications on the public's appreciation of the urban environment. Furthermore, the findings of this study can be considered a starting point for both researchers in the fields of urban conservation, and visual quality of urban streets, whereby the employment of several pictorial analysis techniques for the assessment of visual quality can be further popularised. Moreover, the findings of the study highlight the importance of maintaining a visually pleasing street scenic quality to policy makers such as municipal offices and local city councils. The unfortunate lack of understanding and awareness of the issue of visual quality among the city's public may allow other stakeholders with financial objectives and eased by municipal offices to achieve economic goals, thus increasing the level of visual pollution which contributes to the degradation of the urban scene. Therefore, it is crucial to establish a set of appropriate guidelines and regulations with regard to limiting visual pollution sources to prevent Kuala Lumpur's urban scene being endangered as an attractive multicultural city in Asia. It is worth mentioning that significant differences between subject groups regarding certain levels of environmental stimulation but not others were registered. For instance, it was noticed that for different age groups the older group registered higher levels of sensitivity towards visual pollution. Further detailed investigation on the effect of age on visual comfort is required to fully understand the phenomenon.

**Author Contributions:** Conceptualization, Ammar Al-Sharaa, Mastura Adam, Shamsul Abu Bakar and Norafida Ab Ghafar; methodology, Ammar Al-Sharaa, Mastura Adam and Shamsul Abu Bakar; software, Riyadh Mundher and Ameer Alhasan; formal analysis, Ammar Al-Sharaa and Mastura Adam; investigation, Ammar Al-Sharaa, Shamsul Abu Bakar and Mastura Adam; data curation, Ameer Alhasan, Riyadh Mundher and Norafida Ab Ghafar; writing—original draft preparation, Ammar Al-Sharaa, Mastura Adam and Norafida Ab Ghafar; writing—review and editing, Mastura Adam, Norafida Ab Ghafar and Ameer Alhasan; visualization, Ammar Al-Sharaa, Ameer Alhasan and Riyadh Mundher; supervision, Shamsul Abu Bakar, Mastura Adam, Norafida Ab Ghafar and Ameer Alhasan; project administration, Shamsul Abu Bakar and Mastura Adam. All authors have read and agreed to the published version of the manuscript.

**Funding:** This research received no external funding.

**Data Availability Statement:** Not applicable.

**Conflicts of Interest:** The authors declare no conflict of interest.

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
