# Peer review of "The Effects of Colour Content and Cumulative Area of Outdoor Advertisement Billboards on the Visual Quality of Urban Streets"

_ijgi, doi:10.3390/ijgi11120630_

Round 1
Reviewer 1 Report
The paper wants to investigate the topic of visual pollution, particularly of outdoor advertisements in the urban landscape, and how the color content of outdoor billboard advertisement design can affect the public’s appreciation of urban environments.
The study utilizes a pictorial survey, R.G.B bivariate histogram technique, and an areal cumulative analysis of pictures collected along one of main Kuala Lumpur's commercial and touristic streets.
The methods of the research are well described, the research is original and significant, with an interesting scientific approach. I just suggest a better clarification if the used methods are original by the authors or consolidated in the reference scientific literature:
167: "two pictorial analysis techniques, namely pictorial RGB content analysis and, billboard areal analysis" Could you please add any references to explain if these methods are consolidated or, on the contrary, originally conceptualized by the authors?
174: "Figure 4 shows the images used for investigating": Figure actually shows the photo shooting points, not the images
A second issue regards the Respondents’ Demographic Analysis (273): a homogeneous sample with little age difference (age variation from 20 years to 34 years old), and most of the subjects are students. Maybe it would have been more interesting to have a more diverse sample that also included workers, unemployed, and retirees. The research admits the need for further investigation, these further considerations could be added to the conclusion.
Reviewer 2 Report
only professional proofreading is required before publishing.
Reviewer 3 Report
Many thanks for the opportunity to review the manuscript “The Effects of Colour Content and Cumulative Area of Outdoor Advertisement Billboards on Urban Scenes” submitted to the jornal “IJGI”, MDPI. The manuscript presented a good structure and organization in terms of overall quality. Some comments were provided to increase the value before the final decision.
The title should be different, following some steps like: ask questions, be clear and informative and include only the essential information. In contrast, the abstract makes sense. The author included the main topics like context, objective, methods, results and contributions.
Also, the key-words have attention to the acronyms like “RGB” and repeated words from the title. Please use only relevant words to your research avoiding that are too specific.
Following the introduction, the authors must separate the Introduction and Literature Review/Background. Please consider selecting the topic “1.1 Colour in the Urban Environment and Visual Comfort” as Literature Review/Background section number 2. Keep the introduction with four paragraphs: 1) 1st theorical contextualization; 2) 2nd and 3rd connection between theory and your study, purpose of your study; 3) 4th paragraph contributions, questions, hypothesis, objective… In addition, show the significant contribution to the literature: explain the state of the knowledge with references; identify the gap in the knowledge to fill with the study; outline primary and/or secondary objectives; and outline question(s)/problems if possible.
However, the main part of the study is Methodology. The authors did a great job on this topic. They used active voice; described the study site; presented the methods and statistics in the same order of research/ described in detail the criteria, test, analyses, and techniques. Anyone should be able to replicate this study with the data and information provided by the authors.
Then, the authors describe the results avoiding commentary and interpretation. They gave us an opportunity to understand the method present in the previous section. The illustrations were appropriated, showing the results without repeating the information from materials & methods. One suggestion, highlight the impact of the author's results.
Finally, discussion and conclusion could be one topic named “Discussion and Conclusion” where the authors should review starting with recap of your main findings, put the results in perspective with other reports in the literature, explain significant of results, and how they contribute to the overall state of knowledge, or how they advance knowledge. Outline strengths and limitations. Emphasis on the implications and future studies.
Congratulations to the authors for the quality of the study. I wish you success in the future working in another research.
Reviewer 4 Report
Spiegare meglio le motivazioni e l'applicazione delle analisi effettuate.
Non è ben spiegato come è stata applicata la metodologia (test di correlazione di Pearson e analisi ANOVA della varianza) e in risposta a quale domanda di ricerca.
Pertanto, non è possibile capire se sia opportuno applicare questi metodi rispetto ad altri.
In English: Better explain the reasons and the application of the analyzes carried out.
It is not well explained how the methodology (Pearson's correlation test and ANOVA analysis of variance) was applied and in response to what research question. Therefore, it is not possible to understand whether it is appropriate to apply these methods over others.
